# Anti-Incretin Gut Features Induced by Feed Supplementation with Alpha-Amylase: Studies on EPI Pigs

**DOI:** 10.3390/ijms242216177

**Published:** 2023-11-10

**Authors:** Kateryna Pierzynowska, Piotr Wychowański, Kamil Zaworski, Jarosław Woliński, Janine Donaldson, Stefan Pierzynowski

**Affiliations:** 1Department of Biology, Lund University, 223 62 Lund, Sweden; stefan.pierzynowski@biol.lu.se; 2Department of Animal Physiology, The Kielanowski Institute of Animal Physiology and Nutrition, Polish Academy of Sciences, 05-110 Jabłonna, Poland; k.zaworski@ifzz.pl (K.Z.); j.wolinski@ifzz.pl (J.W.); 3Anara AB, 231 32 Trelleborg, Sweden; piotr.wychowanski@unicatt.it (P.W.); janine.donaldson@wits.ac.za (J.D.); 4Department of Head and Neck and Sensory Organs, Division of Oral Surgery and Implantology, Institute of Clinical Dentistry, Gemelli Foundation for the University Policlinic, Catholic University of the “Sacred Heart”, 00168 Rome, Italy; 5Department of Oral Surgery, Medical University of Gdańsk, 80-211 Gdańsk, Poland; 6Specialized Private Implantology Clinic Wychowanski Stomatologia, 02-517 Warsaw, Poland; 7Large Animal Models Laboratory, The Kielanowski Institute of Animal Physiology and Nutrition, Polish Academy of Sciences, 05-110 Jabłonna, Poland; 8School of Physiology, Faculty of Health Sciences, University of the Witwatersrand (WITS), Johannesburg 2050, South Africa; 9Department of Medical Biology, Institute of Rural Health, 20-090 Lublin, Poland

**Keywords:** EPI pigs, amylase, anti-incretin, OGTT, GLP-1, GIP, glucose metabolism

## Abstract

The acini-islet-acinar (AIA) axis concept justifies the anatomical placement of the Langerhans islets within the exocrine pancreatic parenchyma and explains the existence of the pancreas as a single organ. Amylase has been suggested to play a key role as an anti-incretin factor. Oral glucose tolerance tests (OGTT) were performed on 18 piglets in both a healthy (prior to pancreatic duct ligation (PDL) surgery, study Day 10) and an exocrine pancreatic insufficient (EPI) state (30 days after PDL, study Day 48)). Amylase (4000 units/feeding) or Creon^®^ (100,000 units/feeding) was administered to pigs with the morning and evening meals, according to study design randomization, for 37 days following the first OGTT. Blood glucose levels, as well as plasma levels of insulin, GLP-1, and GIP, were measured, and the HOMA-IR index was calculated. EPI status did not affect the area under the curve (AUC) of insulin release, fasting insulin levels, or the HOMA-IR index, while amylase supplementation led to a significant (*p* < 0.05) decrease in the above-mentioned parameters. At the same time, EPI led to a significant (*p* < 0.05) increase in GLP-1 levels, and neither amylase nor Creon^®^ supplementation had any effects on this EPI-related increase. Fasting plasma levels of GIP were not affected by EPI; however, the GIP response in EPI and Amylase-treated EPI animals was significantly lower (*p* < 0.05) when compared to that of the intact, healthy pigs. Orally administered amylase induces gut anti-incretin action, normalizing glucose homeostasis and reducing HOMA-IR as a long-term outcome, thus lowering the risk of diabetes type II development. Amylase has long-lasting anti-incretin effects, and one could consider the existence of a long-lasting gut memory for amylase, which decreases hyperinsulinemia and hyperglycemia for up to 16 h after the last exposure of the gut to amylase.

## 1. Introduction

The role of pancreatic enzymes in the regulation of metabolism remains the subject of a contemporary debate from the time when S. Rothman, a pioneer in exploring the exocrine pancreas in the early 1970s, said, “Pancreatic trypsin should also be recognized as a hormone”. As early as 1976, Arnesjo and Lundquist observed a decrease in insulin release and the improvement of glucose tolerance in rats with long-term CCK-PZ, achieved via stimulated pancreatic enzyme secretion [1]. The first attempts to show the importance of pancreatic enzymes for glucose metabolism were made in 1983 by Schneeman et al. in their study on rats, where the authors showed decreased levels of serum amylase activity in obese animals when compared to lean ones [2]. The first clinical study assessing the effects of pancreatic enzymes on glucose regulation was performed by Kondo et al. and showed similar results [3].

During the last decade, the number of articles describing the role and importance of pancreatic alpha-amylase for both glucose metabolism and the absorption of carbohydrates in the gut has been constantly growing. Nakajima and co-authors reported on the association of low serum amylase levels with the development of metabolic syndrome and diabetes in asymptomatic subjects [4]. One year later, the same research group reported the latent association between low serum amylase levels and insulin resistance in middle-aged adults [5]. Afterwards, research groups worldwide began to investigate the correlations between serum amylase activity, amylase expression, amylase gene copy numbers, and overall metabolic state [6,7,8,9,10,11]. A comprehensive review of amylase in insulin-resistant conditions was performed by Nakajima [12].

Despite the unquestionable importance of the above-mentioned findings, our understanding of the effects of alpha-amylase and its role in glucose metabolism and carbohydrate absorption still remains limited, despite the idea of the insulin-pancreatic acinar axis introduced by Williams and Goldfine [13]. In fact, low levels of amylase were explained by insufficient insulin action on pancreatic acini, and thus, low amylase levels were considered a consequence of impaired insulin production. 

Previous studies from our lab suggested that amylase clearly has important regulatory functions in glucose homeostasis [14,15]. We have also demonstrated that alpha-amylase, applied either intravenously or enterally, exhibits anti-incretin activity by reducing insulin secretion during an intravenous glucose tolerance test (IGTT) (for review, see [16]). Asanuma-Date et al. showed the participation of amylase in glucose assimilation regulation [17], while Date et al. [18] confirmed the ability of amylase to downregulate GLUT2 and inhibit gut glucose absorption. Recently, Zhou et al. [19] demonstrated the dose-dependent ability of microbial amylase to regulate glucose transport. The above-mentioned metabolic features of amylase also form the background of the so-called “miracle” bariatric surgery, which abruptly eliminates diabetes type 2 (DT2) (for review, see [16]). Overall, previous results suggest that oral alpha-amylase, parallel to its starch digestion function, reduces insulin resistance, which is reflected in HOMA-IR index changes.

Thus, alpha-amylase, a well-known component of the commonly used pancreatic enzyme replacement therapy (PERT), has far more functions than simply that of starch digestion to maltose, including various metabolic regulatory features, such as the maintenance of physiological glucose homeostasis and the simultaneous reduction in insulin secretion, which undoubtedly protects pancreatic beta cells from exhaustion—the AIA concept [20]. As a long-term outcome, amylase may reduce insulin resistance [14], thus protecting the host from the development of diabetes type 2 (DT2) and, ultimately, DT1. Recently, it was found that alpha-amylase may induce the proliferation and differentiation of small intestine epithelial cells [21], and at the same time, low serum levels of pancreatic amylase are significantly associated with colorectal cancer and adenoma in non-alcohol drinkers [22]. Moreover, alpha-amylase has been found to be an important energy regulator in the brain, whose activity is reduced with the development of amyloid beta pathology [23,24]. It is worth pointing out that alpha-amylase is present in mother’s milk and is secreted in large amounts during the early stages of postnatal life from the pancreas and salivary glands. Since mother’s milk does not contain any starch, from a digestion point of view, alpha-amylase would not have much to do. Moreover, glycogen, which is known as “animal starch”, is resistant to the actions of alpha-amylase. Amylase is unable to digest glycogen due to its inability to attack the branching (1→6) bonds. During the neonatal period, and probably even later on in life, amylase could be a factor that inhibits carcinogenesis, e.g., for neuroblastoma [25] in infants. 

The present study was designed to highlight the existence of gut “memory” or gut-born reflexes in response to enteral amylase, as well as the commonly overlooked role of amylase in the regulation of glucose homeostasis and insulin secretion. Additionally, we aimed to confirm the unique features of amylase in acting against hyperglycemia and hyperinsulinemia, which distinguishes amylase from other pancreatic enzymes.

## 2. Results

### 2.1. Glucose Absorption

Comparisons of glucose levels during the OGTT in healthy pigs fed the HFD (healthy control) and exocrine pancreas insufficient (EPI) pigs fed the HFD (EPI control) or the HFD with Creon or with amylase, respectively, are presented in Figure 1A–D. 

Glucose absorption AUCs in Control EPI pigs were significantly lower than those in Control Healthy pigs and Creon-treated EPI pigs (*p* < 0.05). There was only a tendency (*p* = 0.097) for lower glucose absorption observed in Amylase-treated EPI pigs. Creon treatment significantly increased AUCs in EPI pigs to values similar to those observed before pancreatic duct ligation (PDL). Maximal glucose absorption values (Cmax) were highest in Control Healthy and EPI pigs treated with Creon, compared to the Cmax values observed in Control EPI pigs (no enzyme treatment). Fasting glucose values (baseline values) were similar in all EPI pigs. There was a trend for reduced fasting blood glucose values in Amylase-treated pigs (*p* = 0.0845) compared to Control EPI pigs fed the HFD (Figure 1A–D).

### 2.2. Insulin Release and HOMA-IR Index

Comparisons of insulin release during the OGTT in healthy Control pigs and in EPI pigs fed the HFD are presented in Figure 2A–D. Insulin levels in healthy Control pigs were significantly higher compared to those in Control EPI pigs and in EPI pigs treated with amylase. 

The AUC values of insulin release in Amylase-treated pigs were lower than those observed in Creon-treated EPI pigs. It is worth noticing that Creon-treated EPI pigs exhibited the highest intra-individual variations in insulin release, while in Amylase-treated EPI pigs, the insulin release response to glucose loading was homogenous. 

Cmax values for insulin were significantly lower in Control EPI pigs and Amylase-treated EPI pigs when compared to healthy Control pigs and Creon-treated EPI pigs. The fasting insulin level values were most homogenous in the Amylase-treated EPI pigs, and at the same time, they were significantly lower than those observed in any of the other groups (Figure 2).

HOMA-IR was lower in the Amylase-treated EPI pigs compared to the Creon-treated EPI pigs and healthy Control pigs, implying improved insulin resistance (Figure 3).

### 2.3. Glucagon-Like-Peptide 1 (GLP1) and Glucose-Dependent Insulinotropic Polypeptide/Gastric Inhibitory Polypeptide (GIP)

GLP1 and GIP analyses (Figure 4A–D and Figure 5A–D) revealed interesting results. GLP1 levels (Figure 4A–D) measured during the OGTT and presented as AUCs, Cmax, or fasting levels in healthy pigs were significantly lower than those observed in the EPI and Amylase-treated EPI pigs during the OGTT. GLP1 levels in EPI pigs treated with Creon were not different from those observed in the healthy Control pigs.

On the contrary, GIP values (Figure 5A–D) obtained during the OGTT in Control EPI pigs and in EPI pigs treated with Amylase, presented as AUCs or Cmax levels, were significantly lower than those observed in healthy Control pigs.

### 2.4. Body Weight Gain

Figure 6 shows the weekly body weight gain of pigs from particular groups. EPI pigs did not gain any body weight after PDL surgery, while a significant increase in body weight was observed in pigs treated with either Creon or amylase following PDL.

## 3. Discussion

Insulin resistance and insulin sensitivity are recognized as negative and positive symptoms, respectively, in common clinical practice. However, when considering modern Western eating behavior and common dietary and social habits, the above-mentioned judgment seems to be somewhat unfair. Increased sugar consumption, together with high insulin sensitivity, is dangerous since the classic “couch potato” will convert all the sugars to fat. There is no reference evidence that morbidly obese individuals (over 300 kg b wt.—far beyond Class III obesity) develop diabetes type 2. These individuals most probably belong to the metabolically healthy obese (MHO) class but have cardiovascular diseases (CVD) [26]. The insulin production and insulin sensitivity of such patients are probably dangerously adequate for the number of sugars consumed to convert them to fat.

On the other hand, “regular” obese patients usually develop insulin resistance (hyperinsulinemia), together with hyperglycemia. If the above-mentioned status is not treated quickly enough, it can develop into diabetes type 2 (DT2) with CVD, which sooner or later will be converted to diabetes type 1 (DT1) due to pancreatic beta-cell exhaustion/failure. DT2 and even DT1 are in fact both easily curable, and the health status of classically obese patients is not at all comparable to that of morbidly (beyond Class III obesity) obese patients with no signs of diabetes—MHO individuals. Thus, paradoxical insulin resistance should also be recognized as a protection mechanism acting against the development of obesity. 

The mechanism responsible for this phenomenon could involve the reduction or limitation of the amount of sugar that can be converted to fat because of the development of insulin resistance (hyperinsulinemia), considering the theoretical capacity of the extra insulin produced, to which the body is resistant. This could significantly increase blood glucose levels, resulting in the development of all the associated metabolic syndrome consequences and related illnesses [27]. However, it could also serve as a signal to the patient and their physician that there is an overconsumption of sugars, which should be limited as soon as possible. The weight gain (mostly in fat mass) in obese, insulin-resistant patients is probably not as effective as that in MHO individuals with normal insulin sensitivity. The same bad habit of overeating is undoubtedly dangerous, but the development of insulin resistance gives the patient a chance to start controlling their overconsumption of food, ultimately enabling them to live longer, while the insulin sensitivity in MHO patients leads to a quick, sudden death.

Insulin resistance also serves as a satiety signal, which may explain the fact that MHO patients and terminally obese patients without insulin resistance cannot stop eating since they do not develop insulin resistance. Recent studies from our lab involving bariatric surgery (for review, see 16) undoubtedly show that amylase, which has commonly only been associated with starch digestion, should be considered as a main cofactor responsible for the postprandial appearance of glucose in the blood, thus demonstrating anti-incretin potential. 

The “anti-incretin” theory, which forms the basis of the mechanisms behind successful metabolic surgery, was presented by Rubino et al. [28,29,30] at the beginning of the century. The authors look at the gut as a target to regulate metabolism and describe the balance of “anti-incretins” as a crucial factor that regulates glucose metabolism and could be impaired in metabolic disorders. In their comment to the article by Lindquist et al. [31], describing an increase in beta-cell mass after RYGB surgery, Rubino and Amiel [30] point out that proliferation is not always beneficial and in fact could be due to impaired “incretin/anti-incretin balance”. The same research group demonstrated an improvement in insulin sensitivity in diabetic rats after duodenal-jejunal bypass surgery without changes in incretin or insulin secretion [32] and revealed the anti-incretin effect of orally administered glucose [33]. 

However, no endogenous “anti-incretin” factor has been recognized until now. At the same time, the number of reports revealing the alternative, anti-incretin-like role of alpha-amylase has been constantly increasing over the last few years. 

Results from the current study suggest that amylase clearly “sensitizes” the gut to anti-incretin features, which is evidenced by the para-digestive actions of amylase. The current study (Figure 1 and Figure 2) also highlights the long-lasting anti-incretin properties of amylase, manifested via the regulatory capacity of the gut with regards to glucose metabolism. 

The most intriguing and important finding of the present study was the revelation of the long-lasting (persisting) effects of amylase. After 3 weeks of regular treatment with dietary amylase, strong anti-incretin effects were observed within the gut for up to 16 h after exposure of the gut to amylase. Studies describing the long-lasting ant-incretin features of the gut, related to gut exposure to amylase, are ongoing in our lab. The amylase-taught gut anti-incretin reaction was very different from the gut response to Creon when administered in an identical manner. Creon treatment had rather the opposite effect on amylase, stimulating glucose absorption and insulin release. The observed effects of Creon are most probably a result of the activation of PAR receptors via pancreatic proteases, which in turn increases insulin production and thus the HOMA-IR index. Additionally, the HOMA-IR results presented in Figure 3 also showed that minimal insulin resistance was present in pigs treated with amylase prior to the OGTT.

Finally, GLP1 measurements obtained during the OGTT definitively prove the anti-incretin “knowledge” of the intestine pretreated with amylase. GLP1 levels were highest in EPI pigs during the OGTT. Coincidentally, insulin levels in the same blood samples obtained from the same pigs were the lowest. Thus, GLP1, a potent stimulator of insulin release, was unable to stimulate insulin release during the OGTT in EPI pigs pretreated with dietary amylase for 3 weeks. It is necessary to point out that GLP1 levels were significantly higher in all EPI pigs compared to healthy pigs. The low fasting blood glucose and insulin levels in EPI pigs treated with amylase, even together with a high level of GLP1, could suggest some sort of inhibition or slowing down of diabetes development in individuals with high amylase secretion. However, when considering incretin factor “number 2” (in terms of potency), GIP [34,35], the situation becomes more complicated. EPI- and Amylase-treated pigs had the lowest levels of GIP during the OGTT. Thus, the low levels of GIP originating in the stomach could be the main metabolic contra-partner for amylase to inhibit insulin secretion. All the above proves that amylase acts directly via the ‘sensitization’ of the gut to the actions of anti-incretins, and it should be recognized as a factor that regulates blood glucose levels. 

The body weight curves of pigs obtained during the experiment (Figure 6) reflect the significant growth of the pigs treated with Creon. However, even pigs treated with amylase exhibited similar body weight gain to those treated with Creon. EPI pigs that did not receive any enzyme treatment did not grow following PDL. Hence, the next interesting finding of our study is that the digestive enzyme amylase, commonly not recognized as a growth stimulator, in contrast to lipase or protease, exhibits potent growth-stimulating effects. Future studies on growth quality and body composition are undoubtedly necessary. 

Considering the results described above and earlier reports from our lab, as well as previous studies from different countries, which confirm that low serum amylase activity is coincident with DT2 and obesity [4,5,6,7,8,9,10,11,12], we suggest that a revision of the descriptions/meanings of insulin resistance vs. insulin sensitivity is needed. 

### 3.1. Study Limitations

The findings of this study must be considered in light of some limitations. Firstly, it is a holistic observational study, which cannot clarify the pathways of amylase-gut-incretin interactions. Secondly, the oral glucose tolerance test used in our research could not reflect or determine any mechanisms of intrapancreatic relationships, and the use of an euglycemic clamp would have improved the outcome. Direct measurements of insulin sensitivity in future studies could improve our understanding of intrapancreatic interactions. Moreover, as the biologically active gut-derived substances with the ability to influence insulin release are not limited to GIP and GLP-1 [34], it could be useful to include measurements of other gastrointestinal neuroendocrine peptides with possible incretin function, such as obestatin, adrenomedullin, gastrin, etc., in future studies.

### 3.2. Conclusions and Future Directions

Oral amylase promotes the clear, direct, insulin-independent development of gut memory (an anti-incretin-acting memory that decreases hyperinsulinemia and depresses hyperglycemia). Thus, gut memory to amylase exposure normalizes glucose homeostasis, reducing the risk of the development of diabetes in this way. Therefore, the ratio of the number of pancreatic enzymes (especially the amount of amylase vs. proteinase and lipase) observed in healthy subjects, compared to EPI patients when pancreatic enzyme replacement therapy (PERT) is introduced, could be an important or even governing factor to stop or promote diabetes development. Investigation of the nature (kind of regulatory peptides, neural pathways, or direct amylase action originating in the gut after amylase incorporation into the brush border or into enterocytes) of the observed gut anti-incretin memory is necessary for future studies to establish corresponding clinical guidelines.

## 4. Materials and Methods

### 4.1. Animals

The present study was performed in strict accordance with the recommendations in the Guide for the Care and Use of Laboratory Animals of the National Institutes of Health. All efforts were made to minimize animal suffering. The study was approved by the Second Local Ethics Committee for Animal Experimentation in Warsaw, Poland (approval no. WAW2/025/2022). The experiment was performed on 18 crossbred ((Polish Landrace × Yorkshire) × Hampshire)) pigs (Sus scrofa domesticus) of both genders with a body weight of 15 ± 2.3 at the beginning of the study.

### 4.2. Feed and Enzyme Administration

During the study, pigs were fed a high-fat diet (HFD) (Kcynia, Morawski Plant, Lodz, Poland) in an amount equivalent to 4% of their body weight daily, with 1% given at the morning meal (09:00–10:00 a.m.) and 3% at the afternoon meal (17:00–18:00 p.m.). Upon arrival at the experimental unit housing, the pigs were fed a cereal-based, pelleted, standard diet, which was gradually changed by day 5 to a HFD, containing 17.5% crude protein, 3.9% crude fiber, 20% crude fat, and 5.2% ash, together with 5000 IE/kg vitamin A, 500 IE/kg vitamin D, and 85 mg/kg vitamin E. 

### 4.3. Enzymes

Amylase DS100 (Amano Enzymes, *Aspergillus oryzae*, 4000 units/dose) or Creon 25000 (Abbot USA, 100,000 units/dose) were administered to pigs with the morning and evening meals, depending on experimental design randomization. Randomization was performed after the completion of the adaptation period (day 11) and before pancreatic duct ligation (PDL) surgery. Group I (n = 6) was fed HFD alone; Group II (n = 6) and Group III (n = 6) were fed HFD + Creon (2 × 100,000 units daily) or amylase (2 × 4000 units daily) at the morning (ca 1% b wt.) and evening (ca 3% b wt.) meals, respectively, for the entire duration of the experiment. The above-described division of the daily enzyme supplementation portion ensures almost equal caloric contents of the OGTT with regular morning feeding. A detailed study design with randomization is presented in Figure 7.

### 4.4. Study Flow

At the end of the study, all pigs were euthanized by an intravenous injection of sodium pentobarbiturate (100 mg/kg) (Morbital (Biowet, Puławy, Poland)).

### 4.5. Blood Sampling

The first oral glucose tolerance test (OGTT) was performed on healthy animals prior to pancreatic duct ligation surgery after the termination of the adaptation period (Day 10). The second OGTT was performed after the complete development of EPI (Day 48).

Blood samples during the OGTT were collected via the jugular vein catheter one hour and then again at one minute prior to glucose infusion and then at 5, 15, 30, 45, 60, and 120 min after the oral administration of glucose and transferred to BD Vacutainer^®^ glass Aprotinine K_3_EDTA tubes (BD Diagnostics, Franklin Lakes, NJ, USA). The blood samples were immediately placed on ice before they were centrifuged at 3000× *g* for 15 min at 4 °C, and plasma was separated and stored at −80 °C until further analysis. Blood glucose concentrations were measured directly following blood sampling using a glucometer and test strips (Accu-Chek Aviva, Roche Diagnostics, Mannheim, Germany). Plasma insulin and GLP1 concentrations were measured using a porcine insulin ELISA kit (cat #10-1200-01, Mercodia, Uppsala, Sweden) and a GLP-1 multispecies ELISA kit (cat #BMS2194, Invitrogen, Thermo Fisher Scientific Inc., Waltham, MA, USA) according to the manufacturer’s instructions. Plasma GIP concentration was measured with the pig gastric inhibitory polypeptide (GIP) ELISA kit (cat #RK07393-96, ZellBio, Lonsee, Germany) according to the manufacturer’s instructions.

### 4.6. Statistical Analysis

Statistical analysis was performed on the data generated from this study using the ordinary one-way ANOVA for normally distributed datasets or the Kruskall-Wallis test when the data were not normally distributed. The data distribution was assessed using the Shapiro-Wilk normality test. Outliers within data sets were identified using the ROUT method of regression (Q = 0.05%). All the analyses were carried out using GraphPad Prism 10.0, San Diego, USA. The data were not corrected for multiple comparisons. Differences were considered significant if *p* ≤ 0.05; differences were considered a trend when *p* ≤ 0.1; data with a Gaussian distribution are expressed as mean ± standard deviation (±SD); data with a non-Gaussian distribution are expressed as median ± intraquartile range (±IQR). 

## Figures and Tables

**Figure 1 ijms-24-16177-f001:**
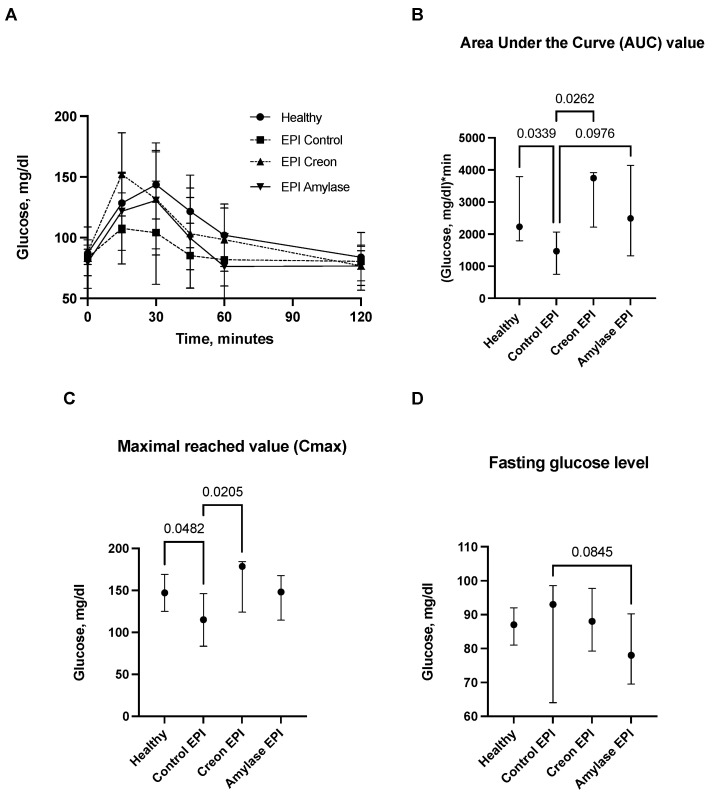
(**A**–**D**) Glucose utilization curves obtained during the OGTT. Control Healthy vs. Control EPI vs. EPI pigs treated with Creon vs. EPI pigs treated with amylase are shown in panel (**A**). The corresponding area under the curves (AUCs), maximal glucose absorption values reached (Cmax), and fasting glucose levels are presented in panels (**B**–**D**). Data on glucose utilization over time are expressed as mean ± standard deviation (±SD); data on AUCs, Cmax, and fasting glucose levels are expressed as median ± intraquartile range (±IQR). Differences between the results were considered significant at *p* < 0.05. A trend for different results was considered when *p*-values ranged between 0.1 and *p* < 0.05. *p* values are given with results.

**Figure 2 ijms-24-16177-f002:**
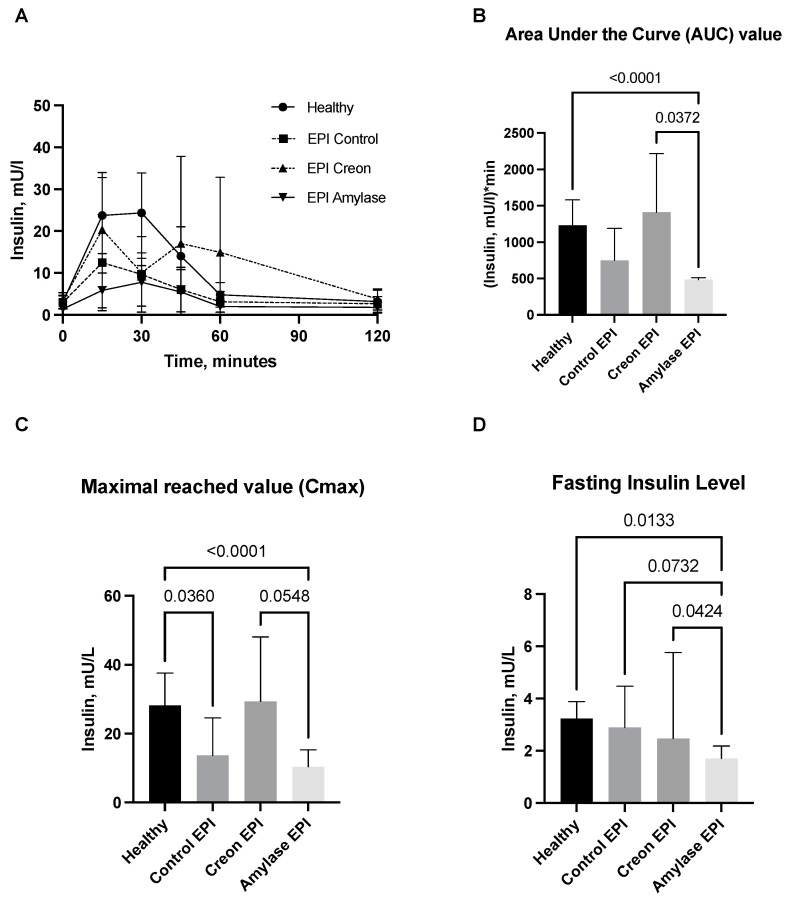
(**A**–**D**) Insulin secretion curves obtained during the OGTT. Control Healthy vs. Control EPI vs. EPI pigs treated with Creon vs. EPI pigs treated with amylase are shown in panel (**A**). The corresponding area under the insulin curves (AUCs), maximal insulin values reached (Cmax), and fasting insulin levels are presented in panels (**B**–**D**). Data on insulin release over time, AUCs, and Cmax are expressed as mean ± standard deviation (±SD); data on fasting insulin levels are expressed as median ± intraquartile range (±IQR). Differences between the results were considered significant at *p* < 0.05. A trend for different results was considered when *p*-values ranged between 0.1 and *p* < 0.05. *p* values are given with results.

**Figure 3 ijms-24-16177-f003:**
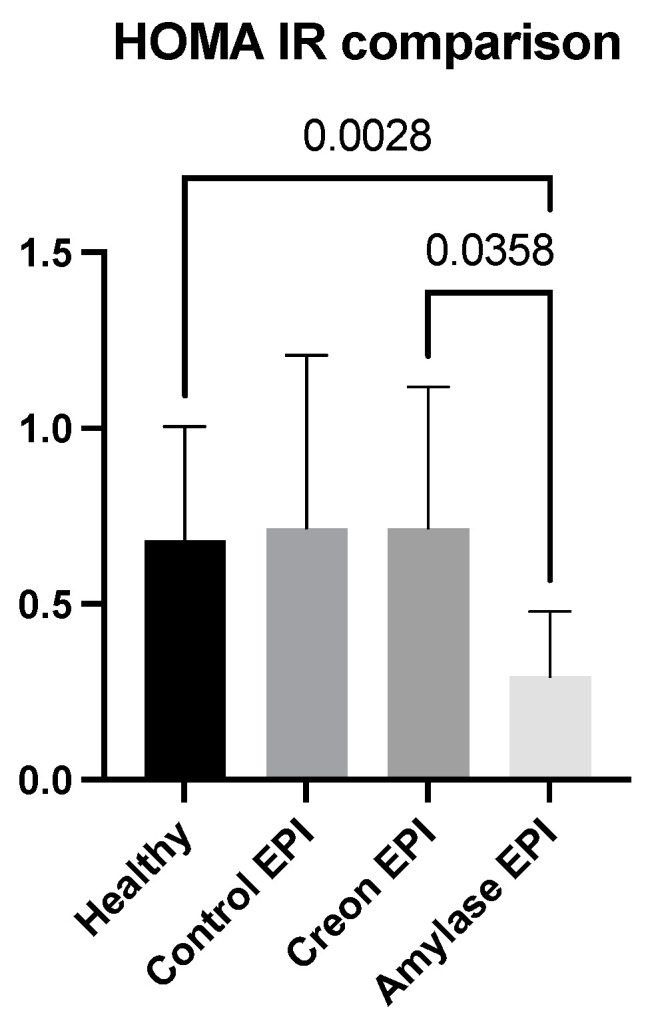
Differences in HOMA-IR obtained when fasting insulin and glucose values from the OGTT were compared. Data on insulin are expressed as mean ± standard deviation (±SD); *p*-values are given with the results bars. Differences were considered significant at *p* < 0.05.

**Figure 4 ijms-24-16177-f004:**
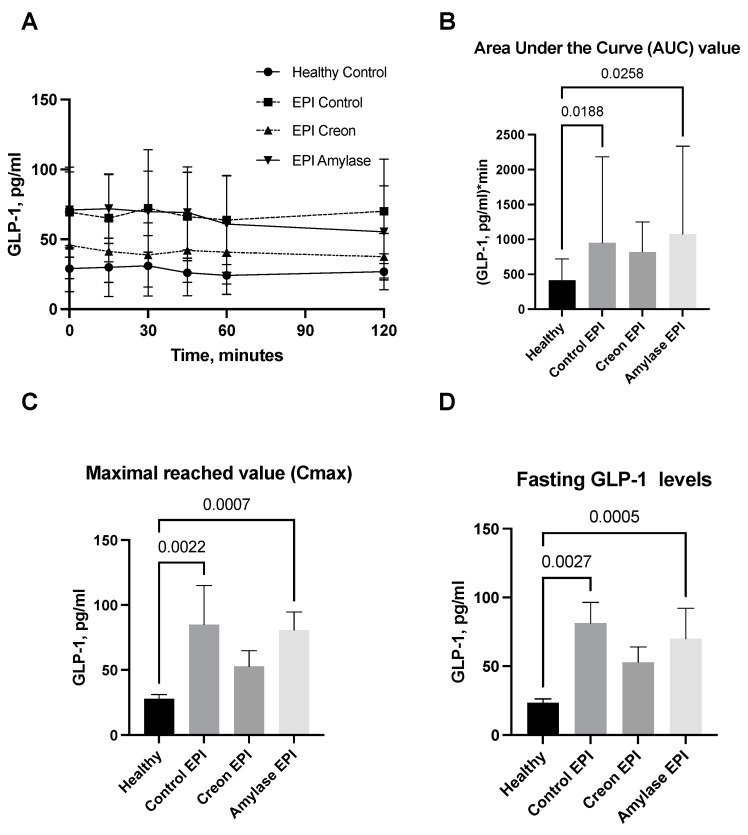
(**A**–**D**) GLP1 levels obtained during the OGTT. The GLP1 secretory curves obtained during the OGTT before EPI development on healthy pigs not treated with enzymes and then during the OGTT after the development of EPI with 3 weeks of either no enzyme treatment or treatment with dietary amylase or Creon are presented in panel (**A**). AUCs, Cmax, and fasting GLP1 values, calculated after deduction of baseline values, are presented in panels (**B**–**D**). Differences between the results were considered significant at *p* < 0.05. A trend for different results was considered when *p*-values ranged between 0.1 and *p* < 0.05. *p* values are given with results.

**Figure 5 ijms-24-16177-f005:**
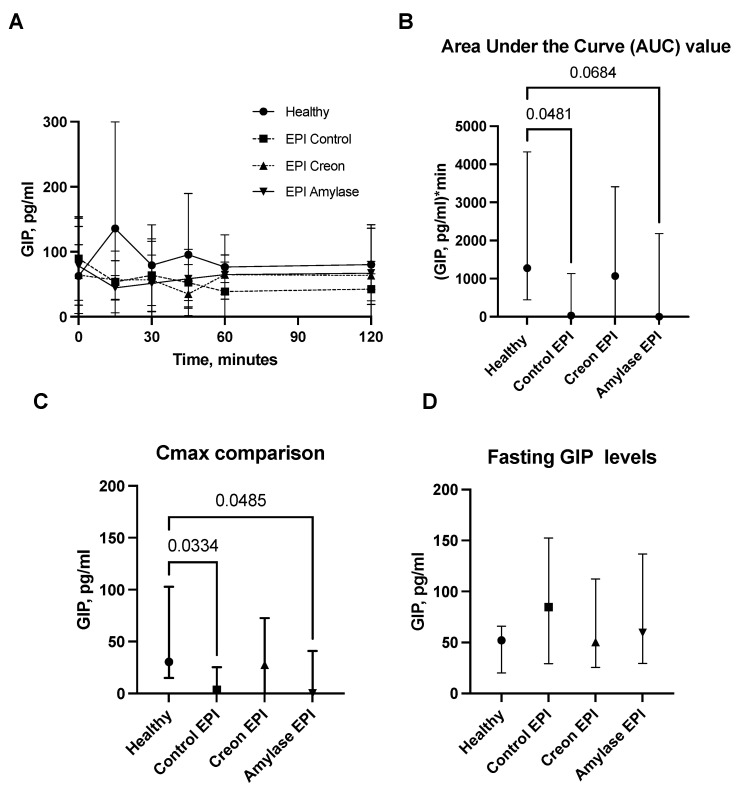
(**A**–**D**) GIP levels obtained during the OGTT. The GIP secretory curve obtained during the OGTT before EPI development on healthy pigs not treated with enzymes and then during the OGTT after the development of EPI with 3 weeks of either no enzyme treatment or treatment with dietary amylase or Creon is presented in panel (**A**). AUCs, Cmax, and fasting GIP values, calculated after deduction of baseline values, are presented in panels (**B**–**D**). Data on GIP secretion over time are expressed as mean ± standard deviation (±SD); data on AUCs, Cmax, and fasting GIP levels are expressed as median ± interquartile range (±IQR). Differences between the results were considered significant at *p* < 0.05. A trend for different results was considered when *p*-values ranged between 0.1 and *p* < 0.05. *p* values are given with results.

**Figure 6 ijms-24-16177-f006:**
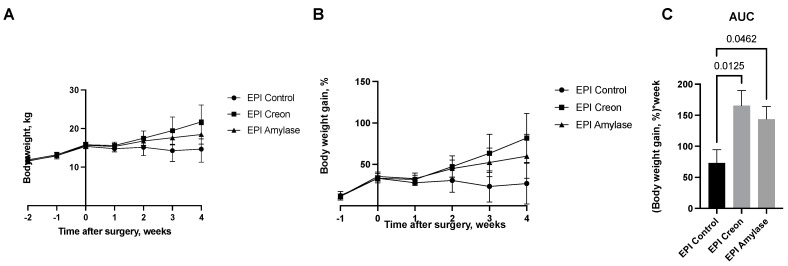
(**A**–**C**) Body weight gain in pigs before and after pancreatic duct ligation (PDL). Data are expressed as mean ± standard deviation (±SD). Differences between the results were considered significant at *p* < 0.05.

**Figure 7 ijms-24-16177-f007:**
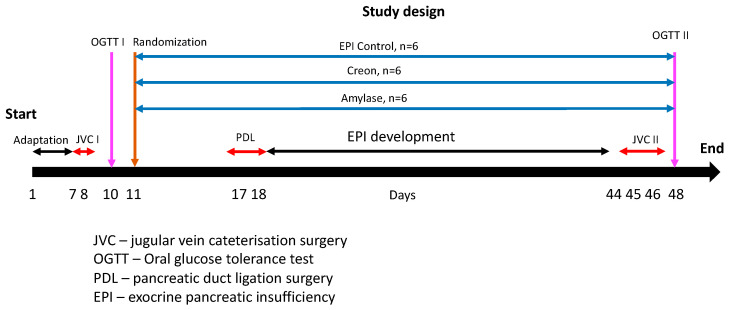
Experiment design and study treatments. Days 1–6—the adaptation period—pigs were successfully adapted to a high-fat diet (HFD). After jugular vein catheter implantation (days 8 and 9), the first OGTT (1 g glucose/kg b wt.) was performed on all pigs following an overnight fast (day 10). On day 11, the pigs were randomized into one of three groups. To introduce meal variation, similar to that of human dietary habits, the pigs were exposed to the following dietary regime: Group I was fed HFD alone; Group II and Group III were fed HFD + Creon (2 × 100,000 units daily) or amylase (2 × 4000 units daily) at the morning (ca 1% b wt.) and evening (ca 3% b wt.) meals, respectively, for the entire duration of the experiment. Pancreatic duct ligation surgery was performed on days 17 and 18. After complete EPI development and second jugular vein catheterization, the second OGTT (1 g glucose/kg b wt.) was performed on all pigs following an overnight fast (day 48).

## Data Availability

The data are contained within the article.

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
