# Peer review of "Anti-Incretin Gut Features Induced by Feed Supplementation with Alpha-Amylase: Studies on EPI Pigs"

_ijms, 2023, doi:10.3390/ijms242216177_

Round 1

Reviewer 1 Report

Comments and Suggestions for Authors

Evaluation of the paper entitled:

Anti-incretin  gut features induced by feed supplementation with alpha-amylase , studies on EPI pigs”

This paper describes the results of experimental study on piglets, being in exocrine pancreatic insufficient state.  These animals were kept on  a high fat diet and treated with amylase ( Amylase study group), or with  pancreatic enzymes (Creon study group), or were untreated (control group). Also healthy pigs were used as normal  control. At the end of experiment oral glucose test was performed on all features. In the pigs treated with amylase  glucose  absorption was a little lower, than in both controls, and Creon group, whereas insulin blood level was markedly decreased, as compared to rest of study groups.  Glucagon-like peptide 1  (GLP1) was significantly higher in pigs with pancreatic insufficiency both ; control and treated with amylase, but GIP was significantly lower in these animals, comparing to healthy control and Creon treated pigs. Application of amylase or Creon to the pigs with pancreatic insufficiency resulted in increased body weight, as compared to controls.

Authors concluded that exposure of animals to the amylase produced anti-incretin effect , normalized glucose homeostasis and thus lowered the risk of diabetes type 2 development.

The results are original and interesting, but I have some comments and suggestions for Authors.

1.       Abstract should be  corrected. Text presented  the experiment inaccurately, the duration of study, doses of amylase used in animals  were not précised,  also the blood levels of glucose, insulin, GLP1, GIP were not shown. In general  - abstract does not describe the results of study. Conclusion should be placed at the end of abstract, not in the middle of its text.

2.       Results: I suggest to remove some of figures and to reduce the number of them  (f.ex. Figs 1 B, C, D, A, 2B, C, D,  4B, C, D , perhaps Figs E) for making the resting figures  bigger and easier to read. In the present form the graphs are unclear, because they are too small.

3.       Methods – the duration of  study should be précised  - which day  the blood was collected ?

4.       Discussion could be shortened.

5.       References : there are too many  selfcitations (about 40 % of all citations). It will be better to include more references from the other authors.

Comments on the Quality of English Language

Author Response

Thank you for the detailed and highly professional evaluation of our manuscript. We hope that our responses and the changes we have introduced into the paper will satisfy you and you will find the edited manuscript acceptable for publication. Please find the detailed responses below and the corresponding corrections could be easily found in track changes in the re-submitted manuscript file.

Comment 1: Abstract should be corrected. Text presented the experiment inaccurately, the duration of study, doses of amylase used in animals were not précised, also the blood levels of glucose, insulin, GLP1, GIP were not shown. In general - abstract does not describe the results of study. Conclusion should be placed at the end of abstract, not in the middle of its text.

Response 1: Thank you for pointing this out. The Abstract has been re-written.

Comment 2: Results: I suggest to remove some of figures and to reduce the number of them (f.ex. Figs 1 B, C, D, A, 2B, C, D, 4B, C, D , perhaps Figs E) for making the resting figures bigger and easier to read. In the present form the graphs are unclear, because they are too small.

Response 2:We agree with this comment. The mentioned Figures have been changed accordingly to the recommendations.

Comment 3: Methods – the duration of study should be précised - which day the blood was collected ?

Response 3:Thank you for pointing this out. The first oral glucose tolerance test (OGTT) was performed on healthy animals prior to pancreatic duct ligation surgery after termination of the adaptation period (Day 10). The second OGTT was performed after the complete development of EPI (Day 48). This explanation was added to the Materials and Methods section of the manuscript.

Comment 4: Discussion could be shortened.

Response 4: Thank you for this comment, we have made an attempt to make discussion section more clear.

Comment 5: References : there are too many selfcitations (about 40 % of all citations). It will be better to include more references from the other authors.

Response 5:We agree with this comment and have included more references from other authors.

Reviewer 2 Report

Comments and Suggestions for Authors

The paper contains sufficiently new and suitable information, and it adheres to the journal’s standards. The topic and level of formality are appropriate for the journal`s readership. Its style and readability are suitable. There is a huge amount of information given throughout the article, but I would suggest revising the paper. 

The authors should add some more suitable keywords.

Article doesn’t demonstrate an adequate understanding of the relevant literature in the field and doesn’t cite an appropriate range of literature sources. References, resource material and literature are poor. I suggest supplementing the Literature Review chapter and add more up to date references.

The methodological concept is clear, and the selected methodology is scientifically appropriate.

The Figure 7- must be better explained.

The Discussion and Conclusion is missing.

The Future directions are missing.

The limitations are missing.

In summary, the article is sufficiently interesting to warrant publication, but it needs major revision. Please follow all the comments above.

Comments on the Quality of English Language

Minor editing of English language required.

Author Response

Thank you for the detailed and highly professional evaluation of our manuscript. We hope that our responses and the changes we have introduced into the paper will satisfy you and you will find the edited manuscript acceptable for publication. Please find the detailed responses below and the corresponding corrections could be easily found in track changes in the re-submitted manuscript file.

Comment 1: The authors should add some more suitable keywords.

Response 1: Thank you for pointing this out. We have included more keywords.

Comment 2: Article doesn’t demonstrate an adequate understanding of the relevant literature in the field and doesn’t cite an appropriate range of literature sources. References, resource material and literature are poor. I suggest supplementing the Literature Review chapter and add more up to date references.

Response 2: Thank you for this suggestion. We have extended the introduction section and modified the discussion section using more up to date references.

Comment 3: The methodological concept is clear, and the selected methodology is scientifically appropriate.

Response 3:Thank you.

Comment 4: The Figure 7- must be better explained.

Response 4: We have introduced more detailed explanation of Figure 7.

Comment 5: The Discussion and Conclusion is missing.

Response 5: We have made an attempt to structure and extend the Discussion section which hopefully makes it clear.

Comment 6: The Future directions are missing.

Response 6:The paragraph has been added

Comment 7: The limitations are missing.

Response 7:The paragraph has been added

Round 2

Reviewer 2 Report

Comments and Suggestions for Authors

I agree with the revised version of the paper. 

Comments on the Quality of English Language

Minor editing of English language required.